# Efficient Adsorption of Methylene Blue by Porous Biochar Derived from Soybean Dreg Using a One-Pot Synthesis Method

**DOI:** 10.3390/molecules26030661

**Published:** 2021-01-27

**Authors:** Zhiwei Ying, Xinwei Chen, He Li, Xinqi Liu, Chi Zhang, Jian Zhang, Guofu Yi

**Affiliations:** 1National Soybean Processing Industry Technology Innovation Center, Beijing Technology and Business University (BTBU), Beijing 100048, China; yingzhiwei0906@163.com (Z.Y.); chenxinweilulu@163.com (X.C.); cauzhangchi@163.com (C.Z.); tsnpzhj@163.com (J.Z.); yiguofu0203@163.com (G.Y.); 2Beijing Advanced Innovation Center for Food Nutrition and Human Health, Beijing Engineering and Technology Research Center of Food Additives, Beijing Technology and Business University (BTBU), Beijing 100048, China

**Keywords:** soybean dreg, porous biochar, methylene blue, adsorption, adsorption mechanism

## Abstract

Soybean dreg is a by-product of soybean products production, with a large consumption in China. Low utilization value leads to random discarding, which is one of the important sources of urban pollution. In this work, porous biochar was synthesized using a one-pot method and potassium bicarbonate (KHCO_3_) with low-cost soybean dreg (SD) powder as the carbon precursor to investigating the adsorption of methylene blue (MB). The prepared samples were characterized with scanning electron microscopy (SEM), transmission electron microscopy (TEM), elemental analyzer (EA), Brunauer-Emmett-Teller (BET), X-ray diffractometer (XRD), Raman spectroscopy (Raman), Fourier transform infrared spectrometer (FTIR), and X-ray photoelectron spectroscopy (XPS). The obtained SDB-K-3 showed a high specific surface area of 1620 m^2^ g^−1^, a large pore volume of 0.7509 cm^3^ g^−1^, and an average pore diameter of 1.859 nm. The results indicated that the maximum adsorption capacity of SDB-K-3 to MB could reach 1273.51 mg g^−1^ at 318 K. The kinetic data were most consistent with the pseudo-second-order model and the adsorption behavior was more suitable for the Langmuir isotherm equation. This study demonstrated that the porous biochar adsorbent can be prepared from soybean dreg by high value utilization, and it could hold significant potential for dye wastewater treatment in the future.

## 1. Introduction

The increasing demand for dyes in different industries exacerbates the discharge of dye wastewater [1], as well as their toxic effect on the environment and organisms, inducing significant concern from society as a whole [2]. For example, the discharge of dye wastewater into forests or fields will directly damage the soil productivity, and the presence of dyes in water can affect light penetration and photosynthesis of aquatic plants [3,4]. Some organic dyes and their products display mutagenicity or carcinogenicity toward humans [5], which may cause different degrees of damage to some organs, including kidney, brain, reproductive system, and liver, etc. [6,7,8]. In the dye industry, the cationic dye MB is an aromatic heterocyclic compound that can burn the eyes of people and animals, and may stimulate the gastrointestinal tract [9], causing symptoms such as dyspnea, vomiting, diarrhea, insanity, and methemoglobin after ingestion [10,11]. Therefore, the effective removal of dye from wastewater and preventing it from polluting the environment is a significant challenge for industrial production that requires an urgent solution.

In the early stage of dye wastewater treatment, only simple balancing and precipitation treatment methods were adopted. Currently, conventional dye removal methods that include biological, chemical, and physical approaches are used for the treatment of dye wastewater [12], such as biodegradation [13], membrane separation [14], coagulation and flocculation [15], photocatalytic degradation [16], and adsorption [17]. Among the above removal methods, adsorption has become one of the preferred technologies for the treatment of dye wastewater because of its low cost, good adsorption, and convenient operation [5]. In the current study, a large number of biomass waste products, such as pineapple crown leaves [18], wood chips [19], bamboo [20], walnut shells [21], bagasse [22], and orange peel [23] are used as raw ingredients in preparing carbon materials that can absorb dye wastewater.

Of all types of biomass waste products, most of them have little or no significant economic value, and the follow-up treatment is also a big problem, especially for the waste with large output such as soybean dregs. China has 5000 years of soybean cultivation history, which is a major oil-bearing crop, as well as one of the important food crops and essential in the human diet [24]. In recent years, China produced more than 80,000 tons of soybean dreg (SD) waste annually. SD is a by-product of the production of soybean products, and since only a small portion of the residue is directed at poultry and livestock feed, the great majority of it is wasted or carelessly discarded in the fields [25]. Processing the considerable amounts of SD produced every year has become a significant challenge in the soybean industry. Furthermore, SD contains about 50% dietary fiber, 25% protein, and 10% lipids and other nutrients [26], which can satisfy the standards of high-value utilization and environmental protection of biomass waste. Moreover, the abundance, low cost, and high fiber content of SD make it an excellent raw ingredient source for the carbon materials.

Currently, the conventional methods of biochar preparation are mainly pyrolysis and hydrogen carbonization [27]. The two methods mainly have two steps: (1) precarbonization of biomass waste and (2) activation of pre-carbonized samples (physical or chemical activation). Comparatively, chemical activation is chosen by more researchers to treat samples, because satisfactory pore structure materials can be obtained, but at the same time, more energy and wastewater are consumed in the production process. The most common activators in chemical activation are KOH, ZnCl_2_, H_3_PO_4_, K_2_CO_3_, and KHCO_3_ [28,29]. Considering the cost of production and environmental sustainability, the one-pot synthesis is an ideal choice.

Considering the economics and feasibility of the adsorbent in practical applications, the regeneration ability and cost economic relevance of the biochars have been studied. The results showed that different desorption mediums had different desorption rate on dye-loaded biochar from different sources, which was mainly due to the different structure of biochar and the adsorption mechanism of dyes. Future work will carry out in-depth studies on the analysis and recycling of biochar adsorbents, so as to improve the practical application of adsorbents.

In this study, porous biochar is prepared using a one-pot method with SD as the carbon precursor and KHCO_3_ as an activator for the adsorption of MB. The structure and composition of the material were characterized by SEM, TEM, EA, BET, XRD, Raman, FTIR, and XPS. The aim of this study was to evaluate the potentiality of the biochar for the removal of MB from aqueous solution and its adsorption mechanism via the study of kinetic and thermodynamic models.

## 2. Results and Discussion

### 2.1. Characterization

#### 2.1.1. SEM and TEM

SEM and TEM characterized the porous structure. According to the SEM image (Figure 1a,b), the surface of the SD appeared as an irregular fold layered structure, which was relatively smooth and dense with no apparent pores [30]. However, after high-temperature activation with KHCO_3_, an abundance of pore structures was evident on the surface of the SDB-K-X, showing a 3D framework with randomly opened pores, indicating that KHCO_3_ played a positive role in the pore formation of biochar. This may be due to the K_2_CO_3_ generated by pyrolysis of KHCO_3_ at 130~170 °C. When the temperature reaches 600 °C, K_2_CO_3_ begins to decompose and expand pores, forming large and abundant pore structures. In addition, metal potassium, reduced by carbon, was converted into gaseous potassium when the activation temperature reached 800 °C, which infiltrated the inner structure of the biochar, forming a larger specific surface area and abundant hierarchically porous structures. The possible activation reactions between SD and KHCO_3_ are as follows (Equations (1)–(5)). SDB-K-3 (Figure 1b) showed the most uniform pore structure, while the pores of SDB-K-2, SDB-K-4, and SDB-K-5 contained blocky impurities, which could be attributed to the different proportions of SD and activators, resulting in an incomplete reaction or skeleton collapse and pore blockage.
2KHCO_3_ → K_2_CO_3_ + CO_2_ + H_2_O(1)
K_2_CO_3_ → K_2_O + CO_2_(2)
K_2_CO_3_ + 2C → 2K + 3CO(3)
K_2_O + C → 2K + CO(4)
CO_2_ + C → 2CO(5)

According to the pore structure analysis with SEM, all of the adsorbents showed continuous three-dimensional pore structure, nanoscile-sized flakes, and fractional porous carbons, but only the internal graphite structure was shown in the high-resolution TEM image of the SDB-K-3 sample (Figure 1c). The pores, indicated by white arrows in Figure 1d, were distributed throughout SDB-K-3, showing distinct lattice fringes with a distance of 0.246 nm, corresponding to the graphite (101) plane. Graphene streaks in TEM images indicated the presence of graphene structures in all samples (Appendix A); however, no clear crystal spacing was detected; this may be attributed to the by-products produced by the reaction between the activator and the carbon in different proportions, which lead to the incomplete or total collapse of the material structure and hinder the exposure of the microcrystalline particles.

#### 2.1.2. EA and BET

The C, O, H, N, and S content of SD and SDB-K-X were measured using an elemental analyzer. The results in Table 1 showed that, after high-temperature activation with KHCO_3_, the biochar yield of the samples exceeded 15%, among which SDB-K-3 reached the highest yield of up to 15.61%. The C content of SDB-K-X increased from 41.14% to over 66%, while the O, H, N, and S content all decreased. The O content of SDB-K-X decreased from 46.63% to about 26% and was probably due to the influence of different KHCO_3_ ratios and the formation of gas during the reaction.

The porous structure of SD and SDB-K-X was characterized by nitrogen sorption isothermal analysis. As illustrated in Figure 2a, SDB-K-X exhibited a typical type I adsorption-desorption isotherm, indicating the microporous structure mainly existed. Figure 2b showed that the pores of SDB-K-X were evenly distributed, and mainly concentrated at 0.75–2.0 nm. With the interaction between substances and the release of gas products in the reaction process, the framework of SDB-K-X was etched to produce a large number of micropores, thus forming a developed layered porous structure [31]. The summary in Table 2 indicates that as the ratio of SD to KHCO_3_ increased, all the SDB-K-X samples displayed extremely high specific surface areas, exceeding 1500 m^2^ g^−1^. When the mass ratio of SD to KHCO_3_ was 1:3, the subsequent SDB-K-3 exhibited the maximum specific surface area and pore volume of up to 1620 m^2^ g^−1^ and 0.7509 cm^3^ g^−1^. As such, the specific surface area first increased and then decreased with the SD to KHCO_3_ ration became higher, which may be ascribed to the shrinking of the carbon framework. This indicates that appropriate amount of KHCO_3_ is conducive to the activation effect; with the increase of KHCO_3_ dosage, the formed micropores will expand into mesopores or macropores. However, when excessive amount of KHCO_3_ is used, some pores will collapse due to excessive corrosion. The analysis was consistent with the SEM and TEM results. In general, materials with a more extensive specific surface area contain more active sites, and a considerable number of micropores are used as the active adsorption sites, which is conducive to the removal of dyes [32].

#### 2.1.3. XRD and Raman

The SDB-K-X were subjected to XRD analysis to characterize the crystalline structures. The SDB-K-X samples all displayed typically disordered amorphous carbon, as suggested by the low-intensity and broadened peaks in the XRD patterns (Figure 2c). There were two main diffraction peaks around 24° and 43°, corresponding to the (002) and (101) plane diffraction (PDF#41-1487), respectively [33].

The Raman spectra (Figure 2d) further confirmed the formation of a graphitized SD structure. Two distinct bands were located at around 1341 cm^−1^ (D band) and 1585 cm^−1^ (G band), providing information regarding the disorder and crystallinity of sp^2^ carbon materials [34]. The I_D_/I_G_ value is usually used to represent the degree of graphitization. All the SDB-K-X samples contained graphitized structures at different degrees, among which the I_D_/I_G_ of SDB-K-3 was the smallest (0.96), indicating that the graphitization degree of SDB-K-3 was relatively high and the structural defects were fewer. This result can be attributed to activators at different quality levels affecting the graphite structure, leading to the formation of a defective texture.

#### 2.1.4. FTIR and XPS

The FTIR spectra of SDB-K-X are illustrated in Figure 3a, indicating that functional groups exist on the surface of biochar, which is different from SD. This may be caused by the volatilization of water and organic matter, as well as the interaction between functional groups during high-temperature activation. The band at 3411 cm^−1^ was ascribed to -OH stretching vibration [21], while the peak at 2933 cm^−1^ was attributed to C-H stretching vibration. Carboxyl C=O stretching vibrations and C=C peaks were identified at 1710 and 1587 cm^−1^; the peak at 1095 cm^−1^ was ascribed to the C-OH stretching vibration [35]. Therefore, the samples contained abundant functional groups, which is conducive to improving the adsorption performance of organic dyes.

Based on the structural characterization analysis of SDB-K-X, XPS was used to evaluate the elemental composition and surface groups of SDB-K-3, which primarily contained C and O, as well as a small amount of N (Figure 3b). These results corresponded well with the EA findings after the peak-differentiating-imitating analysis to identify the high-resolution spectra of C1s and O1s. As shown in Figure 3c,d, the C 1s spectrum was deconvoluted into four peak components at 284.0, 284.8, 286.0, 287.8, and 289.2 eV, which were attributed to C=C (sp^2^C) (1.89%), C-C (sp^3^C) (60.65%), C-O (25.55%), C=O (4.50%), and π-π* satellite (7.40%) groups, respectively [36]. The O1s peaks confirmed the presence of C=O (27.76%), C-O (50.56%), and -OH (21.68%) at 533.7, 532.6, and 531.5 eV [37], respectively. The XPS spectra of SDB-K-2, SDB-K-4, and SDB-K-5 (Appendix A) show similar results. These results were consistent with those of the FTIR characterization, indicating that the SDB-K-X surface contained a considerable number of oxygen-containing functional groups.

### 2.2. The Adsorption Process

Combined with the structural characterization of biochar, MB adsorption onto all the samples was compared in Figure 4; all the samples showed high adsorption efficiency, and SDB-K-3 was selected as the best adsorbent for the removal of MB, which was attributed to its highest specific surface area, suitable pore volume, and abundant oxygen-containing functional groups. The adsorption of SDB-K-3 at different initial concentrations (1000, 1500, and 2000 mg L^−1^) was investigated, indicating that the adsorption capacity increased from 997.85 to 1171.88 mg g^−1^. The adsorption kinetics, adsorption equilibrium, and adsorption thermodynamics of SDB-K-3 for MB were further illustrated in the following sections.

#### 2.2.1. Adsorption Kinetics

To investigate the effect of the sorption behavior mechanism of SDB-K-3 on MB, the kinetic adsorption data were simulated using the pseudo-first-order and pseudo-second-order models. The equations of the two models are expressed as follows [38,39]:(6)$$\ln\left(q_{e}-q_{t\;}\right)={\mathrm{lnq}}_{e}-k_{1}t\;(\mathrm{pseudo}\text{-}\mathrm{first}\text{-}\mathrm{order})$$
(7)$$\frac{t}{q_{t}}=\frac{1}{k_{2}q_{e}^{2}}+\frac{t}{q_{e}}(\mathrm{pseudo}\text{-}\sec\mathrm{ond}\text{-}\mathrm{order})$$
where q_t_ and q_e_ (mg g^−1^) are the amounts of MB adsorbed at time t and equilibrium, respectively, while k_1_ (per min) and k_2_ (g/mg per min) is the pseudo-first-order and pseudo-second-order model rate constants of adsorption, respectively.

Figure 5a shows that the adsorption efficiency increased with the extension of adsorption time, while the adsorption process reached a balance, and the adsorption sites were saturation, maintaining an equilibrium in the adsorption capacity [35]. As the concentration increased, the mass transfer driving force rose, the interaction between SDB-K-3 and MB improved, and the adsorption capacity increased until reaching the adsorption saturation. The maximum absorption Q (mg g^−1^) amounts of 1000, 1500, and 2000 mg L^−1^ MB were 996.37, 1136.83, and 1174.73 mg g^−1^, and the removal efficiency R (%) were 99.64%, 75.79%, and 58.74%, respectively. The adsorption efficiency of SDB-K-3 to MB increased rapidly within 0.5 h and reached the adsorption equilibrium within 2 h. The rapid adsorption process may be ascribed to the existence of effective binding sites, as well as the formation of a hierarchical porous structure with a high SDB-K-3 surface area [40].

The model regarding the relevant kinetic parameters for MB adsorption is provided in Table 3, while Figure 5b,c illustrate the kinetic plots. Therefore, the pseudo-second-order model displayed higher R^2^ values (>0.999) in the three MB concentrations than the pseudo-first-order model. Notably, the pseudo-second-order model can describe the adsorption process better than the pseudo-first-order model, meaning that the chemical effect was also involved in the adsorption of MB on SDB-K-3 [35]. This result can be ascribed to the high surface area and surface abundance of functional groups contributing to the diffusion of organic dye molecules from the water to the surface of the adsorbent. Furthermore, the calculated values of the adsorption amounts at equilibriums of 1000, 1500, and 2000 mg L^−1^ MB onto SDB-6-K from the pseudo-second-order model were 1000.00, 1129.14, and 1155.79 mg g^−1^, respectively, which were approximately the same as those from the experimental results (996.37, 1136.83, and 1174.73 mg g^−1^). These findings suggested that the pseudo-second-order kinetic model was more suitable for describing the adsorption behavior of MB onto SDB-K-3.

#### 2.2.2. Adsorption Equilibrium

The adsorption isotherm is used to describe the interaction between adsorbate and adsorbent, indicating the distribution of adsorption molecules between the liquid and solid phase when the adsorption process reaches an equilibrium state. To analyze the adsorption mechanism of the MB on the surface of SDB-K-3, the adsorption data were fitted using the Langmuir [41] and Freundlich [42] isotherm models, respectively. The Langmuir adsorption isotherm assumes that adsorption occurs in a single-layer at a specific homogeneous site inside the adsorbent. The Freundlich isotherm is an empirical equation suggested for adsorption systems on the heterogeneous surface. The linear form of the Langmuir and Freundlich isotherm equations can be written as:(8)$$\frac{C_{e}}{q_{e}}=\frac{C_{e}}{q_{m}}+\frac{1}{k_{L}q_{m}}$$
(9)$${\mathrm{lnq}}_{e}=lk_{F}+\frac{1}{n_{F}}{\mathrm{lnc}}_{e}n$$
where C_e_ (mg g^−1^) and q_e_ (mg g^−1^) are the concentrations of MB in the solution and the adsorbent at the adsorption equilibrium, respectively. q_m_ (mg g^−1^) is the MB monolayer adsorption capacity, k_L_ is the adsorption constants, which are related to the free energy of adsorption. n_F_ and k_F_ are Freundlich adsorption constants related to adsorption capacity and sorption intensity, respectively.

The adsorption isotherms of SDB-K-3 for MB at different initial dye concentrations and temperatures are shown in Figure 5d. By increasing the initial concentration of MB at a specific temperature, q_e_ gradually increased, and then approached saturation adsorption. With the increase of temperature, the saturation adsorption capacity (qm) increases accordingly, confirming the active influence of temperature on adsorption capacity. Therefore, it can be inferred that the adsorption of SDB-K-3 on MB is an endothermic process.

The curves simulated in Figure 5e, indicate that the curves of the Langmuir model are more suitable for fitting the adsorption data than those of the Freundlich isotherm model (Figure 5f). Therefore, the k_1_ and q_e_ values were calculated from the plots provided in Table 4, indicating that the correlation coefficients (R^2^ > 0.999) of the Langmuir model for all initial MB concentrations were more significant than those found when using the Freundlich model. The results indicated the adsorption process of SDB-K-3 on MB occurred in a single layer.

#### 2.2.3. Thermodynamic Analysis

To estimate the type of adsorption for SDB-K-3 on MB, the thermodynamic adsorption parameters, such as the ΔG (kJ·mol^−1^), the ΔH (kJ·mol^−1^), and the ΔS (J·K^−1^·mol^−1^) were calculated. The thermodynamic parameters can be obtained with the following equations [43]:(10)$${\mathrm{lnK}}_{c}=\frac{-∆H}{\mathrm{RT}}+\frac{∆S}{R}$$
(11)$$∆G=-{\mathrm{RTlnK}}_{C}$$
(12)$$K_{C}=\frac{q_{e}}{C_{e}}$$
where R is the gas constant (8.314 Jmol^−1^ K^−1^), Kc is the thermodynamic equilibrium constant, and T (K) is the sorption temperature, C_e_ (mg g^−1^) and q_e_ (mg g^−1^) are the concentrations of MB in the solution and the adsorbent at the adsorption equilibrium, respectively.

The ΔH and ΔS values can be calculated from the slope and intercept of the linear plot of ln Kc and 1/T. The obtained thermodynamic parameters are shown in Figure 6 and Table 5. All acquired ΔG values were negative at all temperatures, indicating that the adsorption of MB on SDB-K-3 was spontaneous and feasible, suggesting a physisorption process [44]. The positive ΔH values implied the endothermic nature of the adsorption interactions. Furthermore, the positive ΔS value illustrated the adsorbent and the adsorption properties of some structural changes, while indicating the randomness of solid-liquid interface in the process of adsorption was increased [45].

#### 2.2.4. Effect of pH on MB Adsorption

Figure 7 shows the effects of initial pH on MB adsorption by SDB-K-3; the results show that with the increase of the initial pH of MB solution, its adsorption capacity was slightly improved, and the solution pH directly affects the surface charge and the degree of ionization of adsorbents present in the solution [46]. In acidic media, the -COOH and -OH were protonated at the form of -COOH^2+^ and -OH^2+^. The positive charges on the surface of carbon materials compete with MB^+^ to occupy the adsorption sites of the channel, thus reducing the adsorption performance. As the pH of the solution increased, the positive charge on the surface of the adsorbent decreased gradually, which made the electrostatic adsorption between carbon material and MB^+^ occur. When the pH was alkaline, the adsorption capacity of the alkaline dye MB was improved by the -COO^−^ and -OH groups on the surface of SDB-K-3.

### 2.3. Adsorption Performance Comparison with Other Adsorbents

The comparison of the maximum adsorption capacities of MB using various previously reported adsorbents is summarized in Table 6, showing that the adsorption capacity of SDB-K-3 for MB was higher than those of other adsorbents, such as banana peel [46], garlic peel [47], seaweed [48], coconut shells [49], wood [50], rattan [51], and wheat straw [52], which are considered naturally available and cost-efficient. Different adsorbents exhibit different adsorption properties for MB, which may be caused by the variances in raw material composition and preparation methods, resulting in the composition and pore structure of biochar. Consequently, considering its low cost and comparable adsorption capacity, SDB-K-3 is assumed to be a promising adsorption material to remove MB from water.

## 3. Materials and Methods

### 3.1. Materials and Reagents

SD was obtained from the Shandong Yuxin Biotechnology Co., Ltd. (Shandong Province, China). The SD was first dried in a vacuum oven (GZX-9030MBE, Shanghai Boxun Industrial Co., Ltd. Medical Equipment Factor, Shanghai, China) at 105 °C for 48 h to obtain the starting material. All chemicals, including KHCO_3_, MB, and hydrochloric acid (HCl, 37 wt%) were of analytical grade, purchased from the Sinopharm Chemical Reagent Co., Ltd. (Shanghai, China) and used as received.

### 3.2. Preparation of SDB-K-X

The SD was mixed with KHCO_3_ (SD/KHCO_3_ weight ratio = 1:2, 1:3, 1:4, and 1:5) and ground evenly, after which it was activated at 800 °C at a heating rate of 10 °C min^−1^ in a N_2_ (99.999%) atmosphere for 90 min. After the temperature had cooled to room temperature, the subsequent material was placed in a 5 wt% HCl solution and stirred continuously for 24 h, after which it was filtered and separated. The residue was washed with DI water until a neutral pH was reached, and dried at 105 °C for 24 h. The activated material that was obtained was denoted as SDB-K-X (X = 2, 3, 4, and 5 referring to the SD/KHCO_3_ weight ratio = 1:2, 1:3, 1:4, and 1:5, respectively) and finally preserved in a desiccator until further use.

### 3.3. Characterization of SDB-K-X

The yields of biochars were calculated based on a mass balance using Equation (13)
Yields (%) = m1/m2 × 100%(13)
where m1 and m2 were the masses (g) of SDB-K-X and soybean dreg after drying, respectively.

The morphology of SD and SDB-K-X was performed using SEM (JEOL JSM-6700F, Japan) after coating the sample with a thin layer of gold, while the hole wall microstructure was performed with TEM (JEOL JEM-2100F, Tokyo, Japan) operated at 200 kV. The EA (C, H, O, N, and S) of the samples were acquired with an elemental analyzer (Elementar Vario EL III, Hanau, Germany). The nitrogen sorption isotherms of the samples were carried out at 77 K using a nitrogen adsorption apparatus (BET, Quantachrome AUTOSORB IQ, Boynton Beach, FL, USA). The samples were degassed in vacuum at 300 °C for 12 h before testing, the specific surface area and pore size distribution of adsorbents were calculated by density functional theory (DFT) and BET equation. The crystallinity of the samples was conducted with an X-ray diffraction (Rigaku Ultima IV, Tokyo, Japan) using Cu Ka radiation at 40 kV and 40 mA; the scanning rate was 2°/min with a scanning step of 0.2° from 10° to 80° (2θ), while the Raman spectra were obtained using Raman (Bruker Optics SENTERRA, Ettlingen, Germany, excitation-beam wavelength: 532 nm). The organic structure of SDB-K-X was characterized by FTIR (SHIMADZU Type 2000, Tokyo, Japan) recorded in a wavenumber range of 4000–400 cm^−1^ (the sample was thoroughly mixed with KBr at a ratio of 1:100). The surface element composition and chemical state of the samples were obtained with spectra recorded using XPS (Thermo Scientific Escalab 250Xi, Waltham, MA, USA).

### 3.4. Adsorption Experiments

All batches were tested in a Water-bathing Constant Temperature Vibrator (WE-3, Tianjin Ounuo Instrument Co., LTD, Tianjin, China). Then, 0.025 g adsorbent was added to a 25 mL MB aqueous solution of 1000, 1500, and 2000 mg L^−1^, and oscillated at 150 rpm for 2 h at 298 K. The adsorption was performed for 2 h at 150 rpm, after which the suspension was centrifuged at 4000 rpm for 10 min. Then, the supernatant was filtered through a 0.22 μm filter to measure its adsorption value (UV-vis, Agilent Cary-60, Santa Clara, CA, USA) at the maximum value of 664 nm for MB. The adsorption capacity Q (mg g^−1^) and removal efficiency R (%) of MB on the adsorbent were separately calculated with the following formulation:(14)$$Q\left({\mathrm{mg}\;g}^{-1}\right)=\frac{\left(C_{0}-C\right)V}{m}$$
(15)$$R\left(\%\right)=\frac{C_{0}-C}{C_{0}}\times 100\%$$
where Q (mg g^−1^) is the amount of MB adsorbed on the adsorbent, C_0_ (mg L^−1^) and C (mg L^−1^) are the initial concentration and the equilibrium concentration of MB, respectively, V (L) is the volume of the MB solution, m (g) is the mass of the adsorbent used, and R (%) is the removal efficiency of the dye. All adsorption studies were conducted in triplicate, and the data were reported as average ± standard deviation (SD).

#### 3.4.1. Adsorption Kinetics

Furthermore, to investigate the MB adsorption kinetics, 0.025 g of SDB-K-3 was added to 25mL MB aqueous solution at concentrations of 1000, 1500, and 2000 mg L^−1^, respectively, at a temperature of 298 K and a speed of 150 rpm. At a preset time point (from 0 min to 600 min), a small amount of the sample was removed with a pipette gun for determination.

#### 3.4.2. Adsorption Isotherm

The MB of the isothermal adsorption experiments at three different temperatures (298, 308, and 318 K), and 0.025 g SDB-K-3 were added into 25 mL of 250, 500, 750, 1000, 1250, 1500, 1750, and 2000 mg L^−1^ MB, respectively.

#### 3.4.3. Effect of Solution pH

Effect of solution pH was investigated at pH 2, 3, 4, 5, 6, 7, 8, and 9. The pH of solution was adjusted with 1N HCl and 1N NaOH solutions. Then, 0.025 g SDB-K-3 was added to each 25 mL volume of MB aqueous solution having an initial concentration of 2000 mg L^−1^ for a constant adsorption time of 120 min at 150 rpm at 298 K.

## 4. Conclusions

In this study, the porous biochar prepared from SD shows a highly specific BET surface area (1620 m^2^ g^−1^), large pore volume (0.7509 cm^3^ g^−1^), and rich oxygen-containing hydrophobic groups. As a cost-effective dye adsorbent, it has significant application potential. SDB-K-3 exhibits excellent adsorption capacity for MB, with a maximum capability of 1273.51 mg g^−1^ for MB at 318 K. Combined with the detailed characterization of SDB-K-3, this superior result can be ascribed to the 3D framework structure and active adsorption site, suggesting that the combination of chemical adsorption and physical adsorption, π-π interaction, hydrogen bonding, and electrostatic interaction may be possible adsorption mechanisms. These results demonstrate that the excellent adsorption performance of SDB-K-3 to MB is a possible option for practical industrial wastewater treatment. Further investigation should focus on optimizing the preparation method, reducing production costs, and realizing industrial applications.

## Figures and Tables

**Figure 1 molecules-26-00661-f001:**
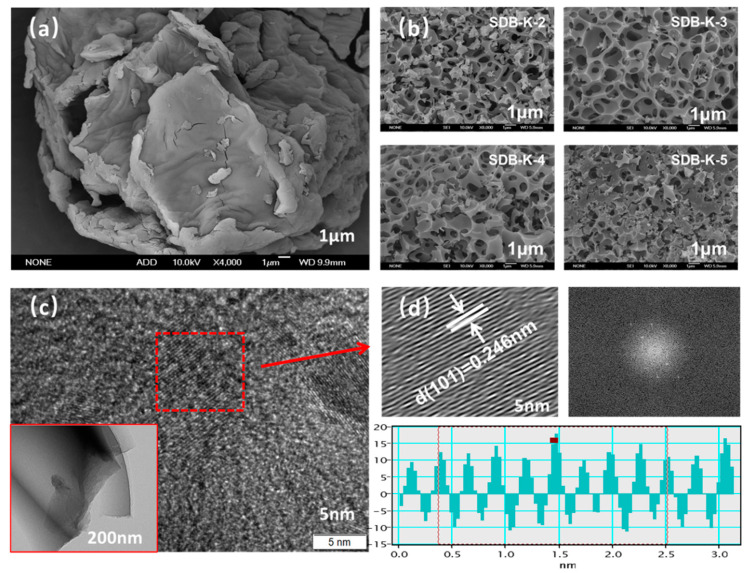
(**a**) SEM images of SD and (**b**) SDB-K-X. (**c**)The TEM image and (**d**) the high-resolution TEM image of SDB-K-3.

**Figure 2 molecules-26-00661-f002:**
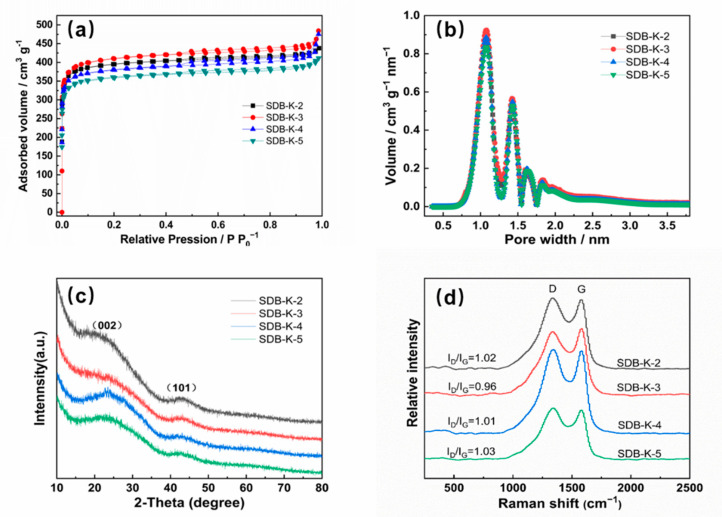
(**a**) The N_2_ adsorption-desorption isotherms and (**b**) pore size distribution of SDB-K-X. (**c**) The XRD image and (**d**) the Raman image of SDB-K-X.

**Figure 3 molecules-26-00661-f003:**
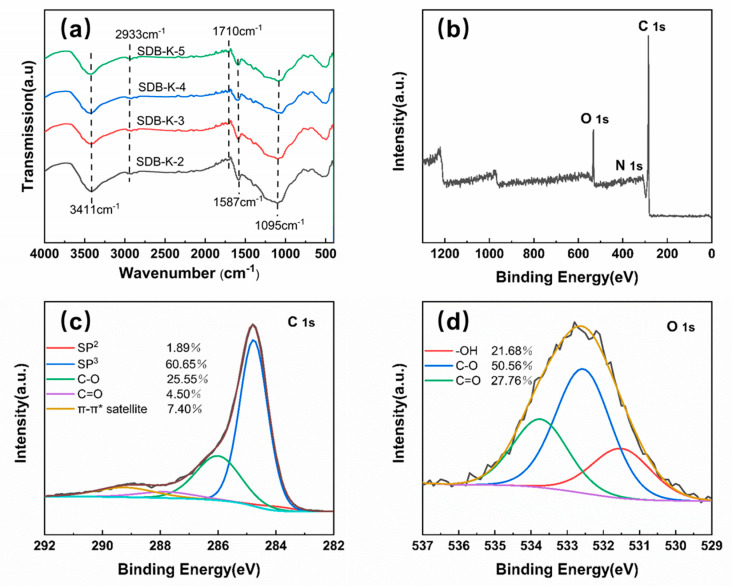
(**a**) FT-IR spectra of SDB-K-3, (**b**) XPS survey, (**c**) high-resolution spectra of C1s, and (**d**) high-resolution spectra of O1s for SDB-K-3.

**Figure 4 molecules-26-00661-f004:**
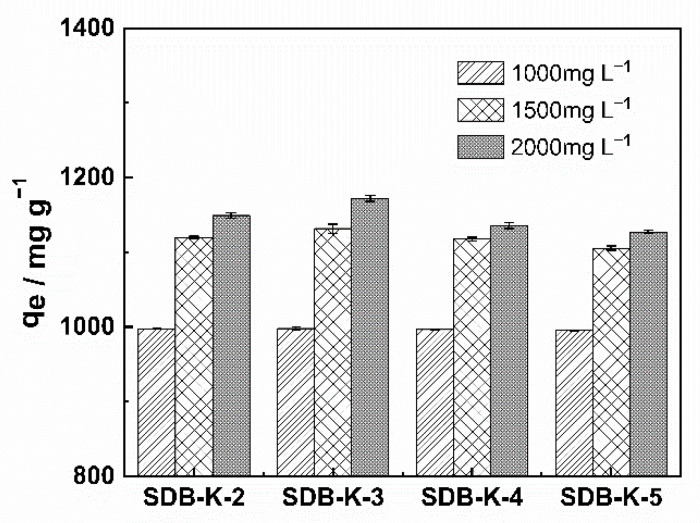
Effect of adsorbent on MB removal.

**Figure 5 molecules-26-00661-f005:**
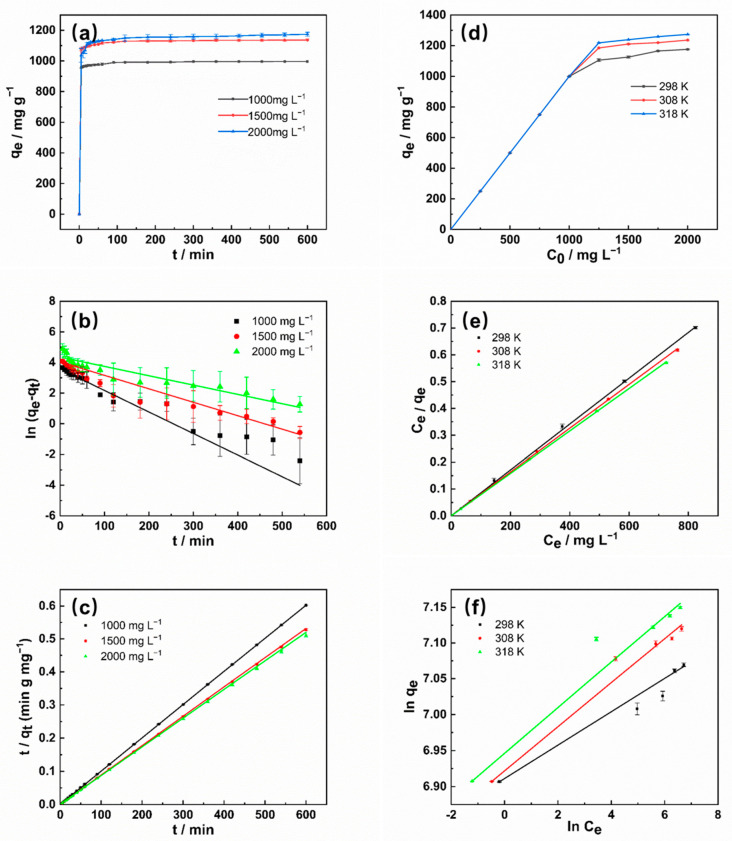
(**a**) The effect of the contact time on MB adsorption by SDB-K-3. Kinetic fitting curves for the MB adsorption by SDB-K-3, (**b**) pseudo-first-order, and (**c**) pseudo-second-order. Conditions: T = 298 K; m = 0.025 g; V = 25 mL; C_0_ = 1000, 1500, and 2000 mg L^−1^. (**d**) The effect of the initial concentration and temperature of the dye on the adsorption performance. (**e**) Linearized Langmuir isotherms for 2000 mg L^−1^ MB adsorption by SDB-K-3 and (**f**) linearized Freundlich isotherms for 2000 mg L^−1^ MB adsorption by SDB-K-3. Conditions: T = 298, 308, and 318 K; m = 0.025 g.

**Figure 6 molecules-26-00661-f006:**
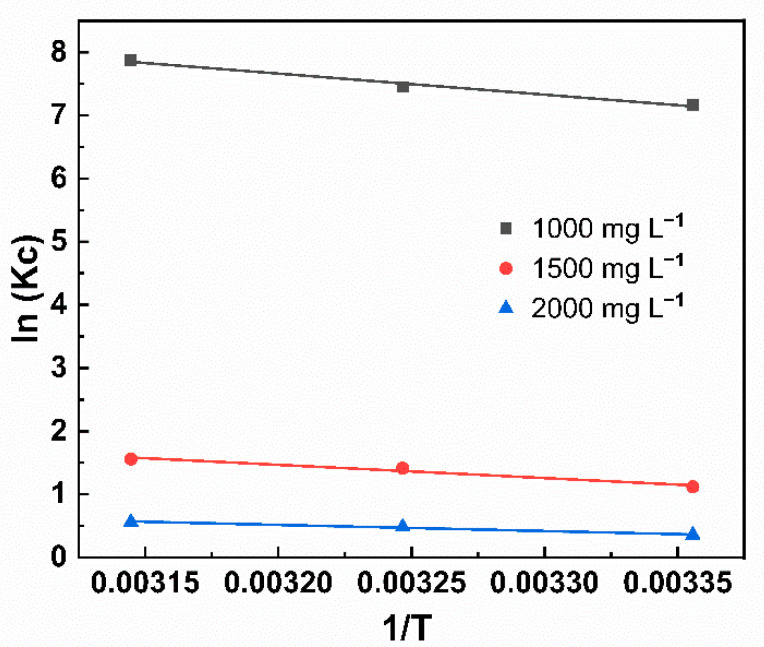
The plot of ln Kc against 1/T for the adsorption of MB onto SDB-K-3. Conditions: T = 298, 308, 318 K; C_0_ = 1000, 1500, and 2000 mg L^−1^.

**Figure 7 molecules-26-00661-f007:**
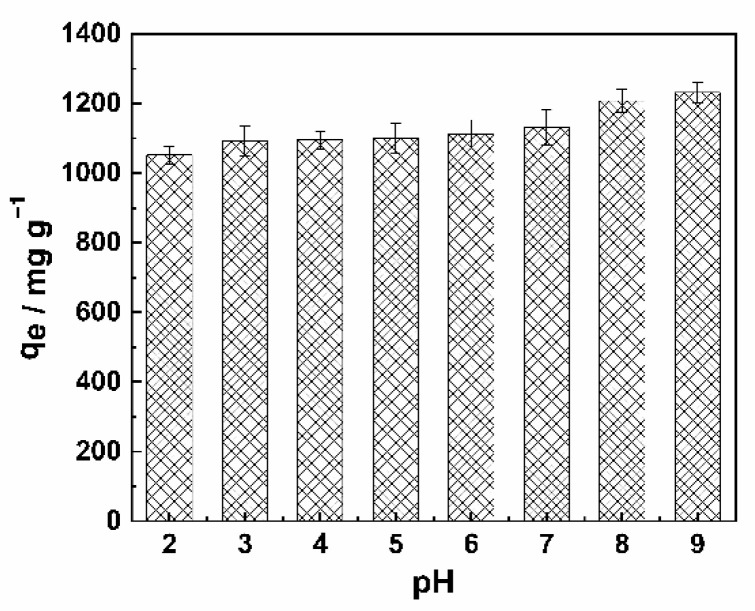
Effect of pH on the MB removal performance by SDB-K-3.

**Table 1 molecules-26-00661-t001:** The yield and the EA of SD and SDB-K-X.

Sample	Yield (%)	C (%)	O (%)	H (%)	N (%)	S (%)
SD	/	41.14	46.63	7.895	3.370	0.476
SDB-K-2	15.34%	66.31	28.27	2.136	0.530	0.348
SDB-K-3	15.61%	66.28	28.11	1.669	0.542	0.383
SDB-K-4	15.11%	69.71	27.81	1.487	0.795	0.528
SDB-K-5	15.00%	69.87	26.84	1.434	0.551	0.445

**Table 2 molecules-26-00661-t002:** Specific surface area, pore volume, and average pore size of SDB-K-X.

Sample	BET (m^2^ g^−1^)	Vtot (cm^3^ g^−1^)	Average Pore Diameter (nm)
SDB-K-2	1572	0.6786	1.727
SDB-K-3	1620	0.7509	1.859
SDB-K-4	1510	0.7375	1.953
SDB-K-5	1425	0.6376	1.790

**Table 3 molecules-26-00661-t003:** Kinetic model parameters for the adsorption of MB on SDB-K-3.

Sample	C_0_ (mg L^−1^)	q_e_, exp (mg g^−1^)	Pseudo-First-Order			Pseudo-Second-Order		
			q_e_,cal (mg·g^−1^)	K_1_ (min^−1^)	R^2^	q_e_,cal (mg·g^−1^)	K_2_ (g·mg^−1^·min^−1^)	R^2^
**SBD-K-3**	1000	996.37	35.74	0.01404	0.87406	1000.00	0.00356	1
	1500	1136.83	56.61	0.00877	0.94358	1129.14	0.00221	0.9999
	2000	1174.73	77.62	0.00608	0.85346	1155.79	0.00086	0.99986

**Table 4 molecules-26-00661-t004:** Isotherm model parameters for the adsorption of MB on SDB-K-3.

Sample	T/K		Langmuir	Freundlich			
		qm (mg·g^−1^)	K_L_ (L·mg^−1^)	R^2^	n	K_F_	R^2^
SBD-K-3	298	1170.72	3.0296	0.99989	43.10	1003.44	0.96722
	308	1225.55	6.65189	0.99992	32.58	1014.14	0.99206
	318	1264.97	11.2932	0.99989	31.45	1039.18	0.99186

**Table 5 molecules-26-00661-t005:** Thermodynamic parameters for the adsorption of MB on SDB-K-3.

Concentration (mg L^−1^)	T (K)	ΔG (kJ·mol^−1^)	ΔH (kJ·mol^−1^)	ΔS (J·K^−1^·mol^−1^)	R^2^
1000	298	−17.70	27.77	152.56	0.96523
	308	−19.22			
	318	−20.75			
1500	298	−2.82	17.41	67.89	0.9409
	308	−3.50			
	318	−4.18			
2000	298	−0.898	8.14	30.33	0.96866
	308	−1.201			
	318	−1.505			

**Table 6 molecules-26-00661-t006:** The comparison of the maximum adsorption capacity of various adsorbents of MB.

Adsorbents	Max Adsorption Capacity qmax (mg g^−^^1^)	Reference
Banana peel	862 mg g^−1^	[46]
Garlic peel	142.86 mg g^−1^	[47]
Seaweed	512.67 mg g^−1^	[48]
Coconut shells	200 mg g^−1^	[49]
Wood	159.89 mg g^−1^	[50]
Rattan	359 mg g^−1^	[51]
Wheat straw	62.5 mg g^−1^	[52]
SD	1273.51 mg g^−1^	Present study

## Data Availability

Not Applicable.

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
