# Peer review of "Efficient Adsorption of Methylene Blue by Porous Biochar Derived from Soybean Dreg Using a One-Pot Synthesis Method"

_molecules, 2021, doi:10.3390/molecules26030661_

Round 1

Reviewer 1 Report

The authors present an experimental study focusing on the synthesis of soybean dreg-based biochar intended for methylene blue adsorption. The manuscript presents several limitations and must be completely improved. The manuscript should not be accepted for publication in Molecules, however, it can be reconsidered if the authors are able to carry out the requested modification.

In general, the manuscript text deserves to be improved, mainly considering the discussion of results. It is clear along the text that the results from material characterisations were only presented by the authors. The authors have to present the relationship between the experimental results and synthesis conditions. The performance of all synthesised must be compared. There is no physical explanation at all.

All material have to be compared based on the analytical characterisation.

The material characterisation section “3.3. Characterization of SDB-K-X” must be entirely improved, taking into consideration that a detailed information of the experimental procedure must be provided for all material characterisations.

English language should be revised in order to correct some typos. For example, “ration” should be replaced by “ratio”.

In the abstract, all acronyms have to be provided, as for instance: SEM, TEM, EA, BET, XRD, Raman, FTIR, XPS and SDB-K-3.

Lattice fringe distance has to be provided for all materials: SDB-K-2, SDB-K-4 and SDB-K-5.

In Table 1, the symbol “%” should be removed from yield values.

Please, correct the title of Y axis in Figure 2b. The Y axis cannot be represented as “volume”. The correct label for the pore size distributions has to be provided.

Figure 2b exhibit multimodal distributions of pore size and the authors have to be aware of the average values arising from multimodal distributions. This point have to be considered in the results discussion.

Considering that all materials exhibited high surface area, the authors must provide adsorption results for all of them as well. Please, provide a full characterisation considering the adsorption performance of all synthesised materials. The performance of the materials must be compared.

Considering that the parameters in Equations 6 and 7 were estimated, the authors have to provide the error associated to the estimated parameters in Table 3. The same must be performed for all synthesised materials.

The same approach must be considered for the results arising from 8 to 12. Without this, the current manuscript seems to have little interest.

The discussion concerning the comparison with other adsorbents must be presented, as the experimental results are discussed in the manuscript text. It does not make sense to reserve a specific topic to do this.

The text associated to “2.4. Analysis of the reusability and cost evaluation” section must be moved to “Introductory section”. Studies focusing on both cost and reusability were not carried out by the authors. The provided information are only descriptive.

Author Response

Response letter

Dear Editors and Reviewers:

Thank you for your kind letter dated November 23, 2020. We thank the reviewers for the time and effort invested in reviewing the previous version of the manuscript. The suggestions enabled us to improve our work. Based on the instructions provided in your letter, we uploaded the files containing the responses to the reviewers and the revised manuscript with all the modifications marked in red.

Appended to this letter is our point-by-point response to the comments raised by the reviewers. The comments are reproduced, and our responses are provided directly below in red.

Supplementary tests were added, the data analysis have been rewritten in response letter and prepared a supplementary file.

Referee #1

  1. Comment: In general, the manuscript text deserves to be improved, mainly considering the discussion of results. It is clear along the text that the results from material characterisations were only presented by the authors. The authors have to present the relationship between the experimental results and synthesis conditions. The performance of all synthesised must be compared. There is no physical explanation at all.

Response: According to your advice, the characterization of the material structure complements the relevant physical explanation in the line in 183-192.

Combined with the structural characterization of biochar, methylene blue adsorption onto all the samples is compared in Figure. 4, all the samples showed high adsorption efficiency, SDB-K-3 was selected as the best adsorbent for the removal of methylene blue, which was attributed to its highest specific surface area, suitable pore volume, and abundant oxygen-containing functional groups. The adsorption of SDB-K-3 at different initial concentrations (1000 mg L-1, 1500 mg L-1, and 2000 mg L-1) was investigated, indicating that the adsorption capacity increased from 997.85 mg g-1 to 1171.88 mg g-1.

Figure 4. Effect of adsorbent on MB removal.

  1. Comment: All material have to be compared based on the analytical characterisation.

Response: The characterization results of missing parts were compared in the analysis in the line 86-138 and 158-181.

2.1.1. SEM and TEM

SEM and TEM characterized the porous structure. According to the SEM image (Figure. 1a, 1b), the surface of the SD appeared as an irregular fold layered structure, which was relatively smooth and dense with no apparent pores [30]. However, after high-temperature activation with KHCO3, an abundance of pore structures was evident on the surface of the SDB-K-X, showing a 3D framework with randomly opened pores, indicating that KHCO3 played a positive role in the pore formation of biochar. This may be due to the K2CO3 generated by pyrolysis of KHCO3 at 130~170℃. When the temperature reaches 600℃, K2CO3 begins to decompose and expand pores, forming large and abundant pore structures. In addition, metal potassium, reduced by carbon, was converted into gaseous potassium when the activation temperature reached 800℃, which infiltrated the inner structure of the biochar, forming a larger specific surface area and abundant hierarchically porous structures. The possible activation reactions between SD and KHCO3 are as follows (eq 1-5). SDB-K-3 (Figure. 1b) showed the most uniform pore structure, while the pores of SDB-K-2, SDB-K-4, and SDB-K-5 contained blocky impurities, which could be attributed to the different proportions of SD and activators, resulting in an incomplete reaction or skeleton collapse and pore blockage.

According to the pore structure analysis with SEM, all of the adsorbents showed continuous three-dimensional pore structure, nanoscile-sized flakes, and fractional porous carbons, but only the internal graphite structure was shown in the high-resolution TEM image of the SDB-K-3 sample (Figure.1c). The pores, indicated by white arrows in Figure. 1d were distributed throughout SDB-K-3, showing distinct lattice fringes with a distance of 0.246 nm, corresponding to the graphite (101) plane. Graphene streaks in TEM images indicated the presence of graphene structures in all samples(Figure S1), however, no clear crystal spacing was detected, this may be attributed to the by-products produced by the reaction between the activator and the carbon in different proportions, which lead to the incomplete or collapse of the material structure and hinder the exposure of the microcrystalline particles.

Figure 1. (a) SEM images of SD and (b) SDB-K-X. (c)The TEM image and (d) the high-resolution TEM image of SDB-K-3.

Figure S1. (a) TEM images of the SDB-K-2; (b) SDB-K-4; (c)SDB-K-5.

2.1.2. EA and BET

The porous structure of SD and SDB-K-X was characterized by nitrogen sorption isothermal analysis. As illustrated in Figure. 2a, SDB-K-X exhibited a typical type I adsorption-desorption isotherm, indicating the microporous structure mainly existsed. Figure. 2b showed that the pores of SDB-K-X were evenly distributed, and mainly concentrated at 0.75-2.0nm. With the interaction between substances and the release of gas products in the reaction process, the framework of SDB-K-X was etched to produce a large number of micropores, thus forming a developed layered porous structure [31]. The summary in Table 2 indicates that as the ratio of SD to KHCO3 increased, all the SDB-K-X samples displayed extremely high specific surface areas, exceeding 1500 m2 g-1. When the mass ratio of SD to KHCO3 was 1:3, the subsequent SDB-K-3 exhibited the maximum specific surface area and pore volume of up to 1620 m2 g−1 and 0.7509 cm3 g-1. As such, the specific surface area first increased and then decreased with the SD to KHCO3 ration became higher, which may be ascribed to the shrinking of the carbon framework. This indicates that appropriate amount of KHCO3 is conducive to the activation effect, with the increase of KHCO3 dosage, the formed micropores will expand into mesopores or macropores. However, when excessive amount of KHCO3 is used, some pores will collapse due to excessive corrosion. The analysis was consistent with the SEM and TEM results. In general, materials with a more extensive specific surface area contain more active sites, and a considerable number of micropores are used as the active adsorption sites, which is conducive to the removal of dyes [32].

2.1.4. FTIR and XPS

Based on the structural characterization analysis of SDB-K-X, XPS was used to evaluate the elemental composition and surface groups of SDB-K-3, which primarily contained C and O, as well as a small amount of N (Figure.3b). These results corresponded well with the EA findings after the peak-differentiating-imitating analysis to identify the high-resolution spectra of C1s and O1s. As shown in Figure.3c, and Figure. 3d, the C 1s spectrum was deconvoluted into four peak components at 284.0 eV, 284.8 eV, 286.0 eV, 287.8 eV and 289.2 eV, which were attributed to C=C (sp2C) (1.89%), C-C (sp3C) (60.65%), C-O (25.55%), C=O (4.50%) and π-π* satellite (7.40%) groups, respectively [36]. The O1s peaks confirmed the presence of C=O (27.76%), C-O (50.56%), and -OH (21.68%) at 533.7eV, 532.6 eV, and 531.5 eV [37], respectively. The XPS spectra of SDB-K-2, SDB-K-4 and SDB-K-5 (Figure 2S.) show similar results. These results were consistent with those of the FTIR characterization, indicating that the SDB-K-X surface contained a considerable number of oxygen-containing functional groups.

Figure 3. (a) FT-IR spectra of SDB-K-3, (b) XPS survey, (c) high-resolution spectra of C1s, and (d) high-resolution spectra of O1s for SDB-K-3.

Figure S2. (a) XPS survey of SDB-K-2, (b) high-resolution spectra of C1s of SDB-K-2, and (c) high-resolution spectra of O1s for SDB-K-2. (d) XPS survey of SDB-K-4, (e) high-resolution spectra of C1s of SDB-K-4, and (f) high-resolution spectra of O1s for SDB-K-4. (g) XPS survey of SDB-K-5, (h) high-resolution spectra of C1s of SDB-K-5, and (i) high-resolution spectra of O1s for SDB-K-5.

  1. Comment: The material characterisation section “3.3. Characterization of SDB-K-X” must be entirely improved, taking into consideration that a detailed information of the experimental procedure must be provided for all material characterisations.

Response: The detailed information of the experimental procedure has been provided for all material characterizations in the line 318-335 in red.

The yields of biochars were calculated based on a mass balance using Equation (13)

Yields (%) =m1/m2× 100%         (13)

where m1and m2 were the masses (g) of SDB-K-X and soybean dreg after drying, respectively.

The morphology of SD and SDB-K-X was performed using SEM (JEOL JSM-6700F, Japan) after coating the sample with a thin layer of gold, while the hole wall microstructure was performed with TEM (JEOL JEM-2100F, Japan) operated at 200 kV. The EA (C, H, O, N, and S) of the samples were acquired with an elemental analyzer (Elementar Vario EL III, Germany). The nitrogen sorption isotherms of the samples were carried out at 77 K using a nitrogen adsorption apparatus (BET, Quantachrome AUTOSORB IQ, USA). The samples were degassed in vacuum at 300℃ for 12 h before testing, the specific surface area and pore size distribution of adsorbents were calculated by density functional theory (DFT) and BET equation. The crystallinity of the samples were conducted with an X-ray diffraction (Rigaku Ultima IV, Japan) using CuKa radiation at 40 kV and 40 mA, the scanning rate was 2◦/min with a scanning step of 0.2◦ from 10◦to 80◦(2θ), while the Raman spectra were obtained using Raman (Bruker Optics SENTERRA, Germany, excitation-beam wavelength: 532 nm). The organic structure of SDB-K-X was characterized by FTIR (SHIMADZU Type 2000, Japan) recorded in a wavenumber range of 4000-400 cm-1 (the sample was thoroughly mixed with KBr at a ratio of 1:100). The surface element composition and chemical state of the samples were obtained with spectra recorded using XPS (Thermo Scientific Escalab 250Xi, USA).

  1. Comment: Lattice fringe distance has to be provided for all materials: SDB-K-2, SDB-K-4 and SDB-K-5.

   Response: TEM apparent analysis of the other three adsorbents: SDB-K-2, SDB-K-4 and SDB-K-5 were performed, and the reasons for the structural differences were analyzed in the line 101-110 in red.

According to the pore structure analysis with SEM, all of the adsorbents showed continuous three-dimensional pore structure, nanoscile-sized flakes, and fractional porous carbons, but only the internal graphite structure was shown in the high-resolution TEM image of the SDB-K-3 sample (Figure.1c). The pores, indicated by white arrows in Figure. 1d were distributed throughout SDB-K-3, showing distinct lattice fringes with a distance of 0.246 nm, corresponding to the graphite (101) plane. Graphene streaks in TEM images indicate the presence of graphene structures in all samples, however, no clear crystal spacing was detected in the other three samples(Figure S1), this may be attributed to the by-products produced by the reaction between the activator and the carbon in different proportions, which lead to the incomplete or collapse of the material structure and hinder the exposure of the microcrystalline particles.

Figure 1. (a) SEM images of SD and (b) SDB-K-X. (c)The TEM image and (d) the high-resolution TEM image of SDB-K-3.

Figure S1. (a) TEM images of the SDB-K-2; (b) SDB-K-4; (c)SDB-K-5.

  1. Comment: Figure 2b exhibit multimodal distributions of pore size and the authors have to be aware of the average values arising from multimodal distributions. This point have to be considered in the results discussion.

Response: I-type (rapid adsorption with a steep curve at the relative low pressure

of 0-0.1) isothermal curves were obtained for SDB-K-X, respectively. This result indicates that SDB-K-X was mainly composed of micropores. This conclusion was confirmed by the pore size distribution (Figure. 2b). SDB-K-X mainly concentrated at 0.75-2.0 nm, the specific surface area, pore volume and the average pore diameter data have been summarized in Table 2. We discussed the results in the line 121-138 in red.

The porous structure of SD and SDB-K-X was characterized by nitrogen sorption isothermal analysis. As illustrated in Figure. 2a, SDB-K-X exhibited a typical type I adsorption-desorption isotherm, indicating the microporous structure mainly existed. Figure. 2b showed that the pores of SDB-K-X were evenly distributed, and mainly concentrated at 0.75-2.0nm. With the interaction between substances and the release of gas products in the reaction process, the framework of SDB-K-X was etched to produce a large number of micropores, thus forming a developed layered porous structure [31]. The summary in Table 2 indicates that as the ratio of SD to KHCO3 increased, all the SDB-K-X samples displayed extremely high specific surface areas, exceeding 1500 m2 g-1. When the mass ratio of SD to KHCO3 was 1:3, the subsequent SDB-K-3 exhibited the maximum specific surface area and pore volume of up to 1620 m2 g−1 and 0.7509 cm3 g-1. As such, the specific surface area first increased and then decreased with the SD to KHCO3 ration became higher, which may be ascribed to the shrinking of the carbon framework. This indicates that appropriate amount of KHCO3 is conducive to the activation effect, with the increase of KHCO3 dosage, the formed micropores will expand into mesopores or macropores. However, when excessive amount of KHCO3 is used, some pores will collapse due to excessive corrosion. The analysis was consistent with the SEM and TEM results. In general, materials with a more extensive specific surface area contain more active sites, and a considerable number of micropores are used as the active adsorption sites, which is conducive to the removal of dyes [32].

Table 2. Specific surface area, pore volume, and average pore size of SDB-K-X.

Sample

BET (m2 g-1

V tot (cm3 g-1

Average pore diameter (nm)

SDB-K-2

1572

0.6786

1.727

SDB-K-3

1620

0.7509

1.859

SDB-K-4

1510

0.7375

1.953

SDB-K-5

1425

0.6376

1.790

  1. Comment: Considering that all materials exhibited high surface area, the authors must provide adsorption results for all of them as well. Please, provide a full characterisation considering the adsorption performance of all synthesised materials. The performance of the materials must be compared.

Response: All the samples were used as adsorbents to carry out adsorption experiments on MB solutions of different concentrations (1000 mg L-1, 1500 mg L-1 and 2000 mg L-1). The experimental results showed that all the samples had efficient adsorption effect. Combined with the structure characterization and the adsorption capacity, SDB-K-3 was determined as the best adsorbent in the later experiments for kinetic and thermodynamic studies.

Figure 4. Effect of adsorbent on MB removal.

  1. Comment: Considering that the parameters in Equations 6 and 7 were estimated, the authors have to provide the error associated to the estimated parameters in Table 3. The same must be performed for all synthesised materials.

Response: The supplementary experiment was done again, and the data were fitted and the error line was analyzed according to the requirements. Both Table 3 and Figure 4 have been revised.

Table 3. Kinetic model parameters for the adsorption of MB on SDB-K-3.

Sample

C0

mg L-1

qeexp

mg g-1

Pseudo-first-order

Pseudo-second-order

qe,cal

(mg·g−1)

K1

(min−1)

R2

qe,cal

(mg·g−1)

K2

(g·mg−1·min−1)

R2

SBD-K-3

1000

996.37

35.74

0.01404

0.87406

1000.00

0.00356

1

1500

1136.83

56.61

0.00877

0.94358

1129.14

0.00221

0.9999

2000

1174.73

77.62

0.00608

0.85346

1155.79

0.00086

0.99986

Figure 5. (a) The effect of the contact time on MB adsorption by SDB-K-3. Kinetic fitting curves for the MB adsorption by SDB-K-3, (b) pseudo-first-order, and (c) pseudo-second-order. Conditions: T = 298 K; m=0.025 g; V=25 mL; C0=1000 mg L-1, 1500 mg L-1, and 2000 mg L-1. (d) The effect of the initial concentration and temperature of the dye on the adsorption performance. (e) Linearized Langmuir isotherms for 2000 mg L-1 MB adsorption by SDB-K-3 and (f) linearized Freundlich isotherms for 2000 mg L-1 MB adsorption by SDB-K-3. Conditions: T = 298 K, 308 K, and 318 K; m=0.025 g.

  1. Comment: The same approach must be considered for the results arising from 8 to 12. Without this, the current manuscript seems to have little interest.

Response: We completed the supplementary experiment in four weeks, and the data were fitted and the error line was analyzed according to the requirements. Table 4 has been revised.

Table 4. Isotherm model parameters for the adsorption of MB on SDB-K-3.

Sample

T/K

Langmuir

Freundlich

qm(mg·g−1)

KL(L·mg−1)

R2

n

KF

R2

SBD-K-3

298

1170.72

3.0296

0.99989

43.10

1003.44

0.96722

308

1225.55

6.65189

0.99992

32.58

1014.14

0.99206

318

1264.97

11.2932

0.99989

31.45

1039.18

0.99186

  1. Comment: The discussion concerning the comparison with other adsorbents must be presented, as the experimental results are discussed in the manuscript text. It does not make sense to reserve a specific topic to do this.

Response: The comparison of the maximum adsorption capacities of methylene blue using various previously reported adsorbents is summarized in Table 6, showing that the adsorption capacity of SDB-K-3 for MB was higher than those of other adsorbents, such as garlic peel [46], banana peel [47], seaweed [48], coconut shells [49], wood [50], rattan [51] and wheat straw [52], which are considered naturally available and cost-efficient. Different adsorbents exhibit different adsorption properties for MB, which may be caused by the variances in raw material composition and preparation methods, resulting in the composition and pore structure of biochar. Consequently, considering its low cost and comparable adsorption capacity, SDB-K-3 is assumed to be a promising adsorption material to remove MB from water.

Table 6. The comparison of the maximum adsorption capacity of various adsorbents of MB.

Adsorbents

Max adsorption capacity qmaxmg g-1

Reference

Garlic peel

142.86 mg g-1

46

Banana peel

862 mg g-1

47

Seaweed

512.67 mg g-1

48

Coconut shells

200 mg g-1

49

Wood

159.89 mg g-1

50

Rattan

359 mg g-1

51

Wheat straw

62.5 mg g-1

52

SD

1273.51 mg g-1

Present study

  1. Comment: The text associated to “2.4. Analysis of the reusability and cost evaluation” section must be moved to “Introductory section”. Studies focusing on both cost and reusability were not carried out by the authors. The provided information are only descriptive.

Response: The content of “Analysis of the reusability and cost evaluation” has been moved to introduction in the line of 73-78.

Considering the economics and feasibility of the adsorbent in practical applications, the regeneration ability and cost economic relevance of the biochars have been studied. The results showed that different desorption mediums had different desorption rate on dye-loaded biochar from different sources, which was mainly due to the different structure of biochar and the adsorption mechanism of dyes. Future work will carry out in-depth studies on the analysis and recycling of biochar adsorbents, so as to improve the practical application of adsorbents.

Reviewer 2 Report

Manuscript ID molecules-1051341 present a valuable research of the use of soybean dreg biochar for Methylene blue adsorption from water. The manuscript can be published, but requires some additions:

  1. The title should be changed. In current form misleads the reader. The manuscript focuses more on the sorbent characterization and the adsorption process than on the synthesis of soybean dreg biochar.
  2. Remove the acronyms from abstract.
  3. Abbreviations section should be introduced to the manuscript.
  4. In the last paragraph of the Introduction section, please clearly define the aim of the research, not what analytical methods and kinetic equations were used.
  5. Lack of statistical analysis of the obtained research results. How many repetitions of the adsorption process have been carried out. In order to obtain reliable results and to determine the significance of differences between the study variants, a statistical analysis should be carried out using appropriate tests.
  6. Standard deviations should be taken into account, which will allow the evaluation of the variability of the obtained efficiency of the adsorption process. It is not sufficient to give only average values.
  7. The Authors should take into account the influence of the pH value on the adsorption process in the manuscript, as it is a parameter of great importance in this process. This should at least be thoroughly discussed.
  8. Graphical abstract will be useful for readers.

Author Response

Referee#2

  1. Comment: The title should be changed. In current form misleads the reader. The manuscript focuses more on the sorbent characterization and the adsorption process than on the synthesis of soybean dreg biochar.

Response: The title of the manuscript is “Synthesis of soybean dreg biochar using a one-pot method for Methylene blue adsorption from water”, the content of this manuscript mainly includes the preparation and characterization of adsorbent and its application in the removal of methylene blue in water, according to your advice, the title of the manuscript has been changed to “Efficient adsorption of methylene blue by porous biochar derived from soybean dreg using a one-pot synthesis method” in the line 2-4 in red.

  1. Comment: Remove the acronyms from abstract.

Response: Full names have been supplemented in the line 18-21 in red.

The prepared samples were characterized with scanning electron microscopy (SEM), transmission electron microscopy (TEM), elemental analyzer (EA), Brunauer-Emmett-Teller (BET), X-ray diffractometer (XRD), Raman spectroscopy (Raman), Fourier transform infrared spectrometer (FTIR), and X-ray photoelectron spectroscopy (XPS).

  1. Comment: Abbreviations section should be introduced to the manuscript.

Response: Thank you for your advice. The abbreviations have been supplemented in the manuscript. According to published journal format, we supplement and refine the abbreviations in text instead of an abbreviations section.

  1. Comment: In the last paragraph of the Introduction section, please clearly define the aim of the research, not what analytical methods and kinetic equations were used.

Response: The aim of the research has added to the manuscript in line 79-83 in red. In this study, porous biochar is prepared using a one-step method with SD as the

carbon precursor and KHCO3 as an activator for the adsorption of MB. The structure and composition of the material were characterized by SEM, TEM, EA, BET, XRD, Raman, FTIR, and XPS. The aim of this study was to evaluate the potentiality of the biochar for the removal of MB from aqueous solution and its adsorption mechanism via the study of kinetic and thermodynamic models.

  1. Comment: Lack of statistical analysis of the obtained research results. How many repetitions of the adsorption process have been carried out. In order to obtain reliable results and to determine the significance of differences between the study variants, a statistical analysis should be carried out using appropriate tests.

Response: In the last four weeks, the relevant experiments were redone and the error analysis and statistical analysis have been carried out.

Figure 4. Effect of adsorbent on MB removal.

  1. Comment: Standard deviations should be taken into account, which will allow the evaluation of the variability of the obtained efficiency of the adsorption process. It is not sufficient to give only average values.

Response: The experiment has been improved according to your suggestions, and the experimental data has been reorganized and analyzed, and the error analysis has been done.

Figure 4. Effect of adsorbent on MB removal. Table 3. Kinetic model parameters for the adsorption of MB on SDB-K-3.

Sample

C0

mg L-1

qeexp

mg g-1

Pseudo-first-order

Pseudo-second-order

qe,cal

(mg·g−1)

K1

(min−1)

R2

qe,cal

(mg·g−1)

K2

(g·mg−1·min−1)

R2

SBD-K-3

1000

996.37

35.74

0.01404

0.87406

1000.00

0.00356

1

1500

1136.83

56.61

0.00877

0.94358

1129.14

0.00221

0.9999

2000

1174.73

77.62

0.00608

0.85346

1155.79

0.00086

0.99986

Table 4. Isotherm model parameters for the adsorption of MB on SDB-K-3.

Sample

T/K

Langmuir

Freundlich

qm(mg·g−1)

KL(L·mg−1)

R2

n

KF

R2

SBD-K-3

298

1170.72

3.0296

0.99989

43.10

1003.44

0.96722

308

1225.55

6.65189

0.99992

32.58

1014.14

0.99206

318

1264.97

11.2932

0.99989

31.45

1039.18

0.99186

Figure 5. (a) The effect of the contact time on MB adsorption by SDB-K-3. Kinetic fitting curves for the MB adsorption by SDB-K-3, (b) pseudo-first-order, and (c) pseudo-second-order. Conditions: T = 298 K; m=0.025 g; V=25 mL; C0=1000 mg L-1, 1500 mg L-1, and 2000 mg L-1. (d) The effect of the initial concentration and temperature of the dye on the adsorption performance. (e) Linearized Langmuir isotherms for 2000 mg L-1 MB adsorption by SDB-K-3 and (f) linearized Freundlich isotherms for 2000 mg L-1 MB adsorption by SDB-K-3. Conditions: T = 298 K, 308 K, and 318 K; m=0.025 g.

Figure 7. Effect of pH on the MB removal performance by SDB-K-3.

  1. Comment: The Authors should take into account the influence of the pH value on the adsorption process in the manuscript, as it is a parameter of great importance in this process. This should at least be thoroughly discussed.

Response: The effect of pH on the adsorption process has been supplemented, and the relevant data and discussion of the experimental results have been added to the manuscript in line 277-289 and 359-363 in red, standard deviation analysis was also carried out.

2.2.4. Effect of pH on MB adsorption

Figure. 7 shows the effects of initial pH on MB adsorption by SDB-K-3, the results show that with the increase of the initial pH of MB solution, its adsorption capacity was slightly improved, and the solution pH directly affects the surface charge and the degree of ionization of adsorbents present in the solution [47]. In acidic media, the -COOH and -OH were protonated at the form of -COOH2+and -OH2+. The positive charges on the surface of carbon materials compete with MB+ to occupy the adsorption sites of the channel, thus reducing the adsorption performance. As the pH of the solution increased, the positive charge on the surface of the adsorbent decreased gradually, which made the electrostatic adsorption between carbon material and MB+ occur. When the pH was alkaline, the adsorption capacity of the alkaline dye MB was improved by the -COO- and -OH groups on the surface of SDB-K-3.

Figure 7. Effect of pH on the MB removal performance by SDB-K-3.

3.4.3. Effect of solution pH

Effect of solution pH was investigated at pH 2, 3, 4, 5, 6, 7, 8 and 9. The pH of solution was adjusted with 1N HCl and 1N NaOH solutions. 0.025g SDB-K-3 was added to each 25 mL volume of MB aqueous solution having an initial concentration of 2000 mg L-1 for a constant adsorption time of 120 min at 150 rpm at 298 K.

  1. Comment: Graphical abstract will be useful for readers.

Response: Thank you for your advice. The Graphical abstract has been added in the manuscript in line 530-533.

Round 2

Reviewer 2 Report

Thanks to the Authors for the improvement of the manuscript. I am satisfied with the changes made. In my opinion, the manuscript can be publish in present form.

This manuscript is a resubmission of an earlier submission. The following is a list of the peer review reports and author responses from that submission.